# Aggression in Group-Housed Male Mice: A Systematic Review

**DOI:** 10.3390/ani13010143

**Published:** 2022-12-30

**Authors:** Elin M. Weber, Josefina Zidar, Birgit Ewaldsson, Kaisa Askevik, Eva Udén, Emma Svensk, Elin Törnqvist

**Affiliations:** 1Department of Animal Environment and Health, Swedish University of Agricultural Sciences, 532 23 Skara, Sweden; 2Department of Animal Environment and Health, Swedish University of Agricultural Sciences, 750 07 Uppsala, Sweden; 3Department of Animal Science and Technology, AstraZeneca, 431 83 Mölndal, Sweden; 4Swedish 3Rs Center, Swedish Board of Agriculture, 553 29 Jönköping, Sweden; 5Swedish National Committee for the Protection of Animals Used for Scientific Purposes, Swedish Board of Agriculture, 553 29 Jönköping, Sweden; 6Department of Animal Health and Antimicrobial Strategies, National Veterinary Institute (SVA), 751 89 Uppsala, Sweden; 7Institute of Environmental Medicine, Karolinska Institutet, 171 77 Stockholm, Sweden

**Keywords:** male mice, group housing, aggression, animal welfare, environmental enrichment, group formation, housing conditions, resident-intruder, social dominance, wound scoring

## Abstract

**Simple Summary:**

When male mice are kept in groups at animal facilities, aggressive interactions between cage mates are not uncommon. Systematically reviewing previous studies that explored the cause of male mice aggression, we found that studies were disparate, using several different strains, a diverse set of environmental enrichments and different ways of grouping and housing mice, as well as different ways to observe aggression. Understanding the cause of male mice aggression is difficult when researchers use different methods and study designs. Nevertheless, our results suggest that home cage aggression is best studied in home cage environments and not by introducing unfamiliar mice to each other in a novel environment. In addition, while we were able to provide recommendations on how to minimize aggression, our assessment was that there is no universal solution that could be used by all animal facilities. Instead, it is important to realize that aggression is complex and that animal facilities might have to try different possible solutions to find what works best under their specific conditions.

**Abstract:**

Aggression among group-housed male mice is a major animal welfare concern often observed at animal facilities. Studies designed to understand the causes of male mice aggression have used different methodological approaches and have been heterogeneous, using different strains, environmental enrichments, housing conditions, group formations and durations. By conducting a systematic literature review based on 198 observed conclusions from 90 articles, we showed that the methodological approach used to study aggression was relevant for the outcome and suggested that home cage observations were better when studying home cage aggression than tests provoking aggression outside the home cage. The study further revealed that aggression is a complex problem; one solution will not be appropriate for all animal facilities and all research projects. Recommendations were provided on promising tools to minimize aggression, based on the results, which included what type of environmental enrichments could be appropriate and which strains of male mice were less likely to be aggressive.

## 1. Introduction

In recent years, several publications have addressed the challenges of keeping male mice in groups (e.g., [1,2,3,4,5]). Mice are social animals and, according to current legislation and guidelines [6,7,8], mice should be group housed when used in research. However, aggression between male cage mates is one of the main problems in laboratory mouse husbandry, affecting both animal welfare and scientific quality [2].

Some general recommendations on how to prevent and minimize aggression have been listed [3,4,9,10]. These include keeping siblings together or grouping familiar mice before sexual maturity, housing male mice in small groups, transferring nesting material at cage cleaning, avoiding disturbances, handling mice with care and using strains with low level of aggression. However, despite being housed according to the general recommendations, mice sometimes fight when kept in groups, illustrating the complexity of the problem.

Aggression in group-housed male mice has been studied using several different methodological approaches [2]. One such approach is to observe undisturbed groups of mice in their home cages; these are usually referred to as home cage observations. There are also a number of different test protocols used to measure aggression and social dominance, such as the resident intruder test [11], the social defeat test [12], the novel arena social encounter test [13] and the tube test [14]. These types of tests mainly focus on territorial aggression and are designed to induce stress, emotional conflict and frustration, with aggression as a readout [2]. They are commonly used in studies where mice are used as animal models to study aggression, social defeat, deficits in social interaction or mood-related disorders (e.g., [12,15]). In these test situations, mice are removed from the social context in their home cage, placed in a novel cage or test arena, and often exposed to unfamiliar conspecifics in different staged encounters. Another way of measuring aggression is to score wounds on the body of the mice, continuously throughout the study and/or after the mice have been euthanized [16]. Sometimes a combination of home cage observations, different test protocols and wound scoring is used.

Aside from differences in methodology, there is a wide variation in different treatments used to study the effects on aggression, such as group composition, differences between strains, handling and cleaning routines, housing conditions and access to environmental enrichment. Environmental enrichment is used to enhance animal welfare by enabling animals to perform positive natural behaviors and increase their ability to gain certain control over their environment [17]. Transferring nesting material at cage cleaning has been shown to decrease aggression [9]. However, the effects of different structural enrichment items on aggression in male mice remain unclear [17]. For example, different studies using shelter as enrichment report both increased aggression [18,19] and no effect on aggression [20,21,22], while combinations of enrichment items with shelter included have been reported to decrease aggression [23,24].

In general, few experimental studies have addressed the problem of home cage aggression under normal husbandry conditions [5]. This might be another reason why results from experimental studies can be difficult to interpret and implement to daily practice in the lab.

To our knowledge, this is the first systematic review on aggression in group-housed male laboratory mice. In this dataset, data were systematically collected from articles investigating this subject. We also included articles that did not have aggression as the primary outcome, but rather as an additional finding. These articles were selected because they could contain valuable information and might not have been included in previous reviews with a focus on animal welfare. Our aim was to map how the literature in the field support, or do not support, available recommendations on how to prevent aggression in group-housed male mice, and to detect knowledge gaps that ought to be filled. We also wanted to address and describe how aggression has been measured in the literature, since this could influence the possibility of translating outcomes to normal husbandry conditions and contribute to useful recommendations.

## 2. Materials and Methods

### 2.1. Literature Search

We conducted a systematic literature search on the 12th of December 2018 using the online databases Medline, Embase and Web of Science. The search included words in titles, abstracts, author keywords and keywords of scientific publications (henceforth referred to as records) and was performed via the University Library at Karolinska Institutet, Sweden. We used the Preferred Reporting Items for Systematic reviews and Meta-Analyses, PRISMA 2020, statement [25]. Details on the search strings are available in the Appendix A. Details of the systematic review protocol have also been registered in the International Platform of Registered Systematic Review and Meta-analysis Protocols (INPLASY) with registration number INPLASY2022120078.

Additional articles from reference lists of three relevant and recent literature reviews [1,2,9] were also included. Duplicate records were removed so that each reference was only represented once.

### 2.2. Inclusion and Exclusion Criteria

The records were screened in a two-step procedure (Figure 1 and Appendix A). In the first step, titles and abstracts were screened to identify empirical studies on group- or single-housed male mice, investigating aggression, social dominance, wounds, stereotypies, stress, physiological parameters or details on husbandry, such as group size, cage cleaning procedures or enrichment. All records that matched these criteria were preliminarily included and the reports were assessed for eligibility in full text following set inclusion and exclusion criteria (Appendix A). To be included the article had to: (i) be available as full text and written in English; (ii) investigate the effects of group housing of male mice (or male mice and castrated male/female mice); (iii) investigate aggression or social dominance. Studies that did not have a specific aim to measure aggression but still drew conclusions about aggression were included. Studies of wild mice or other rodents than mice were excluded.

### 2.3. Data Extraction

The methodology, treatment, outcome of experiments and additional relevant information were extracted from the included articles. Each experimental outcome will hereafter be referred to as an observation.

Sometimes, more than one observation was extracted from the same article, e.g., if the same study explored how enrichment affected aggression in two different strains, or if one study investigated the effect of both group size and kinship.

The 90 included articles were divided between two authors, who extracted information independently. The data were then verified by the author who had not performed the data extraction. The complete dataset, with all extracted information, is present in Appendix A.

### 2.4. Data Analysis

The author’s original conclusions were extracted from the articles and summarized and have in no way been reanalyzed.

The methods used to assess aggression were divided into four categories: home cage observations, test for aggression, wounding and general observation. General observation refers to studies where no specific method could be identified, but the author had made a comment about aggression in the specific experiment.

The treatments used to investigate the effects on aggression were categorized as follows: strain (different strains, substrains or genetically modified strains), enrichment (enriched/non-enriched conditions or different enrichment items used, as well as cage complexity and resource distribution), time spent in group (amount of time spent in a group), group formation (group size, sibling/non-sibling, communal nesting, weaning age, age at group formation, group composition and applying changes to the group composition), housing condition (cage size, density, cage setup, cleaning procedure, female mice close to cage, lightning paradigm, location, scent and identification method) and other (castration, castrated male or female partner, brain weight and general observations where no specific treatment had been applied).

The enrichment items used were first categorized into five major types, related to which behavioral needs they fulfilled: social (contact, non-contact), occupational (psychological, exercise), physical (cage, accessories), sensory (visual, auditory, other stimuli) and nutritional (delivery, type) [26]. To handle the large variation in enrichment items used and facilitate comparisons within these groups, the enrichments used were then further categorized into the following eight subgroups, according to the supposed purpose of the item: shelter, feed enrichment, nesting material, climbing structure, hiding device, locomotor enrichment, gnawing device and other (Appendix A). Shelter included items specifically designed to function as a shelter (e.g., nest box), whereas hiding device were structures that mice could hide behind, in or under, but not specifically designed for use as a shelter (e.g., tube).

## 3. Results

### 3.1. Search and Study Selection

The database literature search revealed 1420 potentially relevant articles. The reference lists of three recent literature reviews [1,2,9] added another 316, for a total of 1736 articles. After removal of duplicates, 1062 abstracts were screened and 605 articles were preliminarily included. In the full text screening, 90 articles were identified as suitable for inclusion in the systematic review (Figure 1).

### 3.2. Description of Data Set

In total, 198 observations were extracted from the 90 articles (Appendix A). Each observation corresponded to the outcome of an experiment with respect to the effect on aggression in group-housed male mice. In most cases, the observations referred to a specific strain and a specific treatment used to affect aggression. Thus, one article, and even one experiment, could include several observations (Table 1). Therefore, comparisons of the outcome on aggression had to be performed among the observations and not between articles.

Data extracted from the articles were divided into different categories based on different treatments used to study the effect on aggression (Table 1 and Appendix A). The most common treatment used to study effects on aggression was enrichment, followed by time spent in group, strain and factors related to group formation such as group size and kinship of groups. Details of the housing conditions, such as cage size and cleaning procedure, were less commonly studied. Thirty-one articles contained observations from more than one treatment category.

### 3.3. Different Methodological Approaches Used to Study Aggression

The methodological approaches used to study aggression can be divided into three categories: home cage observations, test protocols to measure aggression and scoring of wounds. Home cage observations were most common, alone or in combination with other methodological approaches (Figure 2a, Appendix A). Nineteen articles used methodological approaches from more than one category e.g., home cage observations and wound scoring.

The use of different methodological approaches has changed over time (Figure 2b). Home cage observations, alone or in combination with other approaches, have increased in publications after 2001 as well as the practice of using more than one type of method to study aggression. Contrarily, the use of tests to measure aggression as a single methodological approach has decreased.

When using home cage observations as a single approach to study aggression, observations showed decreased aggression, increased aggression and no effect on aggression (Figure 2c). On the other hand, when using tests to measure aggression outside the home cage and wound scoring, as single approaches to study aggression, decreased aggression was seldom observed, while observations of increased aggression or no effect were more commonly reported (Figure 2c). In the sections describing the effects of different treatments below, the methodological approaches used are not described for each observation. Details can be found in Appendix A.

### 3.4. Details and Description of Housing and Husbandry

Details on cage interiors were included in 70% of the articles published before 2011 and in 80% of the articles published after 2011 (Figure 3). During the period 2001–2018, two-thirds of the articles included information on cleaning routines, compared with only one third of the articles published before 2001. On the contrary, information about kinship—whether groups were put together with male mice from different litters or with siblings—were included in a larger proportion of the publications before 2011, compared with articles published between 2011–2018 (from 75–78% before 2011 to 68% during 2011–2018; Figure 3).

### 3.5. The Effect of Enrichment on Aggression

In many of the articles included in this systematic review, mice were housed in cages with enrichment (Appendix A). However, the results presented in this section only include papers that specifically studied the effects of enrichment on aggression. In these studies, the standard cages may have had a baseline level of enrichment (not only bedding). Thus, the treatment was defined as additional items added to the enriched cages.

Enrichment was the most commonly used treatment to study effects on aggression (Table 1). More than 40 different enrichment items were used in the different articles (Appendix A) and different types of enrichment items were often studied together (Appendix A). The majority of all enrichment items used fell under the major type physical enrichment (38 items in total), including the subgroups nesting material (5 items, 18 observations), shelter (6, 18), hiding device (10, 23), climbing structure (9, 13), locomotor enrichment (1, 10) and other (7, 11). The remaining belonged to nutritional enrichments (4 items in total), including the subgroups feed enrichment (2, 9) and gnawing device (2, 2). Of the eight subgroups, hiding device was the enrichment category that most frequently resulted in decreased aggression (Figure 4). This category also had the highest number of observations, some with only one enrichment item and others where the hiding device was tested in combination with other items.

From all articles that studied enrichment, there were 13 observations of decreased aggression [21,23,24,27,28,29,30,31], 32 where the enrichment did not have any effect [20,21,22,23,29,32,33,34,35,36,37,38,39,40,41] and 15 observations of increased aggression [18,19,22,32,34,37,42,43,44,45,46,47]. In addition, there were 4 observations with different outcomes depending on the method used to measure aggression [48,49,50,51] (Appendix A). More than half of the observations of decreased aggression were based on groups formed from non-sibling mice, or substrains of BALB/c mice. Non-sibling groups and groups with BALB/c mice were commonly used also in the observations reporting no effect on aggression. Conversely, sibling groups were more common among the observations of increased aggression and all observations except two were based on other strains than BALB/c. When looking at experiments that investigated the effect of enrichment in C57BL/6, no effect on aggression was most often observed. In one case there was a decrease in aggression, but increased aggression was never observed (Appendix A).

Cage complexity and resource distribution have also been studied [29,52]. Nadiah et al. (2014) used two connected cages and found that distributing a combination of enrichment items over both cages reduced aggression compared to putting all enrichment items in one cage [29]. Bergmann et al. (1995) studied an enriched setting with one or nine passages to the fodder rack. The enriched condition resulted in an increased proportion of wounded mice and the setting with only one passage to the fodder rack had the highest proportion of injuries [52].

### 3.6. Aggression Observed over Time

Changes in aggression over time have been studied in multiple articles (Table 1). However, the amount of time the mice were observed varied from a couple of days to weeks or months, and were therefore seldom comparable. With no exceptions, a within approach was used where aggression was measured multiple times and each mouse or group was compared with itself. In addition, in these studies, there were often interacting factors, such as enrichment items in the home cage, which sometimes had a direct influence on the results (Appendix A). For example, Haemisch et al. in two publications from 1994, reported increased aggression with time in enriched cages but did not see any effect in non-enriched cages [32,48]. Ambrose and Morton (2000), on the other hand, reported increased aggression with time in both enriched and non-enriched cages [24].

From all articles that studied the impact of time spent in a group, there were nine observations of decreased aggression over time [36,53,54,55,56,57,58,59,60,61], ten with no effect [32,48,59,61,62,63,64,65] and 19 observations of increased aggression [13,18,20,24,32,33,48,57,66,67,68,69,70,71,72] (Appendix A). Aggression as an effect of time has been studied repeatedly in BALB/c [18,20,24,33,36,65,69,70], C57BL/6 [33,61,65] and CD-1 [13,20,67,71] mice. BALB/c and CD-1 were more often observed to become increasingly aggressive with time [13,20,24,33,67,69,70,71].

### 3.7. Strain Differences in Aggression

The most commonly used strains in this dataset of 90 articles were BALB/c (24 articles), C57BL/6 (17 articles) and CD-1 (14 articles) (Appendix A). Other strains were used in ≤7 articles.

Differences in aggression between strains, substrains or genetically modified lines were reported in 34 observations from 26 different articles (Appendix A). This included 22 observations of comparisons between two strains (Table 2). Similar to the complete dataset, BALB/c and C57BL/6 were the most commonly used strains. C57BL/6 was observed to be less aggressive than another strain on six occasions [23,73,74,75] and more aggressive only once [33]. BALB/c was observed to be less aggressive than another strain on five occasions [20,33,73,74,76] and more aggressive than another strain on three occasions [73,74,75]. Six direct comparisons between C57BL/6 and BALB/c were identified; three identified BALB/c as more aggressive [73,74,75], two did not see any difference [30,65] and one identified C57BL/6 as more aggressive [33]. C57BL/6 and BTBR have also been repeatedly compared but no differences in overall aggression were observed [61,77].

Differences in aggression between different substrains of C57BL/6 or BALB/c were studied on two occasions [21,41]. Gaskill et al. (2017) compared six different substrains of C57BL/6 and observed no difference in aggression [41] while Giles et al. (2018) saw more aggression after cage change in BALB/cJ as compared to BALB/cByJ [21].

Differences in aggression between genetically modified strains, compared to the corresponding wildtype, have also been observed [72,80,81,82,83,84,85,86,87]. Out of ten observations, six reported increased aggression in the genetically modified strain [72,83,84,85,86,87]. Martinez-Cue et al. (2005) observed a decrease of aggression in a trisomic strain compared to its disomic control [80] while Trainor et al. (2007) saw no effect on aggression comparing a knockout strain with its wildtype [81]. Lewejohann et al. (2010) observed a decrease in aggression in a homozygous knockout strain but no difference in the heterozygous knockout strain [82]. In addition, Sorensen et al. (2005) saw decreased aggression in a transgenic C57BL/6 strain when housed in mixed groups with wildtype animals [72].

### 3.8. Size and Group Composition

An effect of group size on aggression was identified in twelve observations from ten articles (Table 3). However, the definition of small or large groups tended to vary between articles and, while small groups usually refer to three or sometimes four or five mice, large groups can refer to five mice all the way up to 10–12 or even 20 mice. Five observations saw the least aggression in small groups [54,59,70,88,89] while six observed the least aggression in large groups [42,90,91,92].

In this dataset, 65 articles presented information on kinship, whether groups were formed from sibling or mice from different litters (Figure 3, Appendix A). It was more than twice as common to form groups from non-siblings as compared to siblings: 41 and 16 articles respectively. Only six articles studied the effects of group formation on aggression [58,59,63,64,69,93]. In four articles, no effect on aggression was observed when comparing groups of siblings with groups of mice from different litters [59,63,64,69]. In the remaining two articles, one using wild type CD-1 and the other a transgenic line of CD-1, there was increased aggression in the non-sibling groups [58,93].

Bartolomucci et al. (2004) studied group formation from non-sibling CD-1 mice at weaning or adolescence and observed increased aggression in groups formed from juvenile CD-1 mice [93]. The effects of other early life events, such as weaning age [41,94] and nesting condition (standard or communal) [95], were also studied. The two articles studying weaning age showed contrary results; one observed increased aggression after early weaning [94] while the other observed increased aggression after late weaning [41]. When comparing standard or communal nesting, increased aggression was reported in communal nesting [95].

The effects of regrouping mice were studied in seven articles [41,89,94,96,97,98,99]. The group formation in the experiments varied widely and included single housing at delivery to be grouped later [96], rearranging group-housed mice into new groups [41,94,97,98] and removal of alpha males [99]. Only one of these treatments, breaking up groups of fighting mice into smaller groups with wounded mice only, resulted in decreased aggression [89].

### 3.9. Housing Conditions and Male Mouse Aggression

In this dataset, procedures and factors that related to housing conditions were the least studied category of treatments that could affect aggression (Table 1).

Aggression in relation to different cleaning procedures was studied in four articles [24,69,72,100]. Transferring the mice to a clean cage and moving nesting material to the new cage reduced aggression [69,100]. Placing clean sawdust in a soiled home cage and moving soiled sawdust or enrichment items to the clean cage increased aggression [24,69,100].

Another aspect that relates to housing conditions is the cage size, or density in the cage. Eleven observations from seven articles were related to this aspect: three defined as density [38,75] and eight as cage size [43,55,70,88,101]. All three observations of density, as well as three observations of cage size, showed no effect on aggression [38,43,75,101]. These six observations were made in six different strains. Increased aggression with increasing cage size was observed three times [70,76,88]. Two of these observations were based on BALB/c [70,88]. Contrarily, Poole and Morgan (1976) observed decreased aggression with increasing cage size in LACA/CFW mice, both comparing different groups and from within analysis of the same mice in different-sized cages [55].

Other practices relating to housing conditions have also been studied [36,41,102]. Disturbed lighting and having female mice close to the male mice cage increased aggression [36,102]. Gaskill et al. (2017) studied several parameters of the housing condition and observed increased aggression in mice identified by ear notch as compared to tail tattoo and increased aggression in lavender-scented cages, as well as a difference in aggression between racks. No differences were observed within the same rack [41].

## 4. Discussion

In the articles included in this systematic review, researchers investigated how aggression among group-housed male mice was affected by environmental enrichment, strain, age, housing conditions, group formation and time spent in groups. In all treatment categories, there were observations of decreased aggression, increased aggression, as well as no effect on aggression. When compiling the results, it was, in many cases, not possible to single out a clear effect of a specific treatment because of potential bias presented from other parameters. For example, effects of enrichment in the home cage could be affected by characteristics of the strain tested, the amount of time the mice were grouped together before the study or the length of the study itself. Altogether, this confirmed what has been pointed out in previous review articles on aggression in male mice [1,2]: that the problem is complex and that it is unlikely that one solution will fit every situation. We also found indications that the method used to assess aggression could influence the outcome, further complicating interpretation of results. Still, aggression was decreased when using certain environmental enrichment items and certain strains.

### 4.1. Methods Used for Assessment of Aggressive Behaviour

In the included articles, assessments of aggression among group-housed male mice were performed by observations in the home cage, using tests for aggression outside the home cage, by registration of wounds, or by a combination of these methods. Home cage observations of aggression were used in more than half of the articles and this approach has been more frequently used in recent years (Figure 2b). Studying aggressive behavior in the home cage is likely more relevant for understanding aggression under normal husbandry than using aggression tests outside the home cage, as the fighting observed in the home cage will be linked to social behavior in the group, whereas fighting in a novel environment or with an intruder captures aggressive behavior in relation to those factors.

Many conclusions of aggression in male mice have historically been drawn from assessment of aggressive behavior outside the home cage, using tests such as the resident intruder test and social defeat test (Figure 2b). This approach was used, alone or in combination, in about one third of the articles in the present systematic review (Appendix A). Not surprisingly, decreased aggression was rarely seen as an outcome in articles where the aggression was measured outside the home cage (Figure 2c). The large proportion of increased aggression seen when studying aggression outside the home cage could be a result of the method itself rather than a treatment effect, since the purpose of these tests is to provoke aggression. Aggression assessed in the home cage, on the other hand, resulted in decreased, increased or no difference in aggression, evenly distributed between the included articles. The importance of using reliable methods relevant for assessing aggression in the home cage when developing/evaluating preventative measures for group housing male mice was therefore elucidated in this systematic review. The observed effect of the treatments used to study the effects on aggression varied between articles regarding strain, environmental enrichment and group formation. In some cases, the results were contradictory even when using the same comparison/hypothesis. It is thus possible that the increased aggressive behavior was a result of the aggression test methods rather than true effects of the specific treatment.

### 4.2. Environmental Enrichment

Enrichment is commonly used as a means to improve animal welfare and for refinement of animal use in research [103,104]. However, general recommendations on how enrichment should be used to prevent aggression are not straightforward, as enrichment has been reported to have both positive results, negative results or no impact on aggression [2,17]. This was consistent with the results in this review; in almost half of the observations, enrichment had neither positive nor negative observed effects on aggression. Interestingly, enrichment increased aggression in only about 20% of the observations. It is worth mentioning that some research groups are reluctant to use enrichments because of a strong belief that enrichments will increase aggression. The presented results indicate that the type of enrichment used was important, e.g., hiding devices, feed enrichment and nesting materials, used alone or in combination with other enrichments, were associated with decreased aggression in male mouse groups (Figure 4). These enrichments resulted in increased aggression in only seven out of 50 observations (Appendix A). Enrichment items that mice can interact and work with, such as nesting material, often decrease aggression and are generally included in advice for reducing aggression among male mice [3,4,9,10]. In a study based on data from group-housed male mice in 40 animal facilities by Lidster et al. (2019), transfer of clean and dry nesting material at cage cleaning had a clear positive effect, with reduced aggression [3]. Transfer of nesting material was also perceived as the most effective approach to prevent fighting and aggression in male mouse groups when asking animal technicians and researchers at Swedish universities at workshops and in a survey [4]. The workshop and survey respondents also mentioned adding extra nesting material as efficient. In other species, such as non-human primates, visual barriers were recommended to minimize aggression when housing animals in social groups [105]. It could be that providing hiding devices to allow mice to remove themselves from sight of one another mimics the natural behavior response of fleeing when threatened by a conspecific [2,41], thus decreasing aggression in mice.

In the present review, locomotor enrichment (wheel) and shelter were associated with increased aggression in more than half of the observations. Interestingly, a combination of shelter and wheel were used in five of these nine observations. Mice can defend and monopolize valuable physical enrichment items, and only a limited number of items can fit in a conventional mouse cage. Lower levels of aggression when enrichment items were dispersed was found in both male [29] and female [106] mice. It might be that the shelter and wheel were only accessible for one or a few mice at a time, especially when the wheel was connected to the house, resulting in mice displaying resource aggression. If there were only one entrance, it might also be difficult to escape an aggressive encounter, with the risk of escalating aggression as a consequence. This could explain why increased aggressive behavior was found when these types of enrichments were used.

Interestingly, the results presented herein could indicate that environmental enrichment had a more pronounced positive effect when used in non-sibling male groups. There were no observations of decreased aggression in sibling groups when assessing effects of environmental enrichment. This could be due to the overall lower levels of aggression in groups of siblings. On the contrary, increased aggression in enriched environments was observed for both sibling and non-sibling groups (Appendix A).

Recommendations to prevent aggression that considered housing conditions for mice include transfer of nesting material at cage cleaning [3,4,9,10], avoiding disturbances and handling mice with care [9]. However, there are no clear recommendations for enrichment. This could be due to the wide variety of enrichment items and combinations of different items studied. In the reviewed studies, 42 different enrichment items were used in combination with different strains and group constellations, making it very difficult to interpret results and give clear recommendations. Many parameters of normal husbandry can affect aggression in group-housed laboratory mice. Some, such as enrichment, are often represented in empirical studies. Despite this, systematic evaluation of many enrichment items is lacking. Experienced personnel working with mice on a daily basis learn, through trial and error, what type of enrichments reduce or trigger aggression—but this type of information is rarely collected systematically by researchers. Information on basic husbandry and routines were missing in many articles in the present review. A complete methods section is a prerequisite to be able to repeat experiments, or to implement suggested practices in normal husbandry. The ARRIVE guidelines [107], originally published in 2010 [108], should ensure that studies are reported in enough detail. However, several important parameters were often missing from the reviewed studies.

### 4.3. Strain

A previously published recommendation advises the use of strains with low prevalence of aggression whenever possible [3]. In a study from 2019, Lidster et al. collected significant data on incidents of aggressive behavior in group-housed mice at animal facilities and determined strain to be one of the key factors influencing prevalence of aggression. They concluded that C57BL/6 and BALB/c were the two strains that showed lower prevalence of aggression [3]. Among the herein included articles, similar tendencies were found. For example, C57BL/6 mice were less aggressive than their counterparts in all but one study investigating the difference in aggression between strains. BALB/c mice also showed low levels of aggression when compared to other strains. In fact, BALB/c mice were more aggressive only when compared with C57BL/6, indicating that both these strains showed low levels of aggression. Contrary to these results, C57BL/6 has sometimes been reported as an aggressive strain when doing surveys and workshops at animal facilities [1,4]. Because C57BL/6 is a commonly used strain, it is possible that this conception among survey respondents reflected the large number of C57BL/6 housed at facilities, rather than it being a more aggressive strain. That being said, if animal technicians experience aggression among these strains, this should be taken seriously. It is possible that these strains are less aggressive overall and, under certain conditions, display low levels of aggression, but that aggression can be triggered in other circumstances. When choosing what strain to use for animal experiments, we still know very little about what triggers aggression in the various strains. Empirical studies focused on comparing aggression among strains have shown somewhat inconclusive results (Table 2 and Appendix A). Considering that studies vary in a large number of factors other than strain—differences in number of animals per cage, age, cage size, enrichment items, etc.—it was difficult to interpret results and impossible to draw any major conclusions based on this dataset alone. Researchers must explore how aggressive behavior in various strains is influenced by environmental factors. For example, among the included articles herein, we noted a repeated negative relationship between cage size and aggression in BALB/c, something that could be further explored. Preferably, we would like to see systematic studies comparing multiple strains under varying environmental conditions to help us better understand variations in aggressive behavior and how it is influenced by other factors in group-housed male mice.

### 4.4. Group Formation

General recommendations exist, aimed at preventing and minimizing aggression in group-housed male mice. These are related to group formation, such as housing in small groups formed from siblings kept together or familiar mice grouped before sexual maturity [3,9,10]. Recent publications also suggested that male mice can be grouped with unfamiliar mice at weaning [4], shortly after weaning with one-week age difference between males [109], or before sexual maturity [110]. In the present review, there was only one article evaluating age-related aggression or grouping before and after sexual maturity. This article reported increased aggression in groups formed from unfamiliar mice at weaning, as compared to groups formed from littermates or groups formed from somewhat older unfamiliar mice [93]. However, time spent in groups was also studied in some articles, with a time frame varying from a few days up to 40 weeks (Appendix A). Scattered results were observed, including increased aggression, decreased aggression, and no observed change in aggression, indicating that the incidence of aggressive behavior was influenced by factors other than time spent together.

In the present review, there were inconclusive results on preferable group size for male mice. Decreased aggression was observed in smaller groups with 2–5 mice per group as well as in larger groups of 10–20 mice per group (Table 3). The literature on group size was divided; there were recommendations for 3 mice per group in some cases (e.g., [70]), while the incidence of aggression was significantly higher in groups of three, compared to groups of 4 and 5, in others [3].

### 4.5. Recommendations Based on Results from the Systematic Review

Based on this literature review, a few recommendations stand out as promising tools to minimize aggression among group-housed male mice and should be considered.

Enrichment: Use nesting material and other enrichment items that the mice can manipulate and work with. Use hiding devices where the mice can hide. Be careful when using enrichments that can be monopolized by one or a few individuals Note that environmental enrichment seem to be even more important in non-sibling groups.Strain: Use less aggressive strains whenever possible. This literature review indicates that C57BL/6 and BALB/c show low prevalence of aggression, in line with recently published advice [3].Solutions may vary: Diverging and varying opinions exist between research facilities concerning what works to prevent aggression. Try different solutions and adjust to specific needs in your lab. Be aware that the same solution might not work for all strains and research projects.Methods used to study aggression: Be aware that the method used to study aggression could influence the relevance of the results for home cage aggression. It is therefore important to evaluate the method used to study aggression before drawing conclusions on whether or not the reported results should be implemented in daily practice. Recent studies using home cage observations could provide more reliable information.

### 4.6. Important Notes for Future Research

To facilitate the understanding of home cage aggression, the most relevant approach is to study aggression in groups of male mice using home cage observations, where the social group is kept intact (and not by aggression tests outside the home cage with unfamiliar mice).

It is important to include all data regarding housing and husbandry in publications so that results can be interpreted correctly, and comparisons can be made between studies. Follow the ARRIVE guidelines.

More systematic studies are needed, using various strains for the same treatments and housing conditions and the same methods to study aggression.

## 5. Conclusions

Aggression among group-housed male mice is a major welfare concern. Identifying factors that prevent aggression and thus enable mice to be held in stable social groups will not only minimize pain and suffering but also contribute to a reduction in the number of animals used in research.

The keys to success for male mouse group-housing are multifactorial. There is no one solution that fits all; solutions can and will vary and must be adjusted depending on animal facilities and research areas. Finding the factors that work at a given facility may take time and several factors should be evaluated before resorting to single housing. In this literature review, a number of recommendations have been identified as promising tools to minimize aggression among group-housed male mice. Taken together with other current information from the literature, this study could be an important complement when developing guidelines on how to prevent aggression when male mice are housed in groups.

## Figures and Tables

**Figure 1 animals-13-00143-f001:**
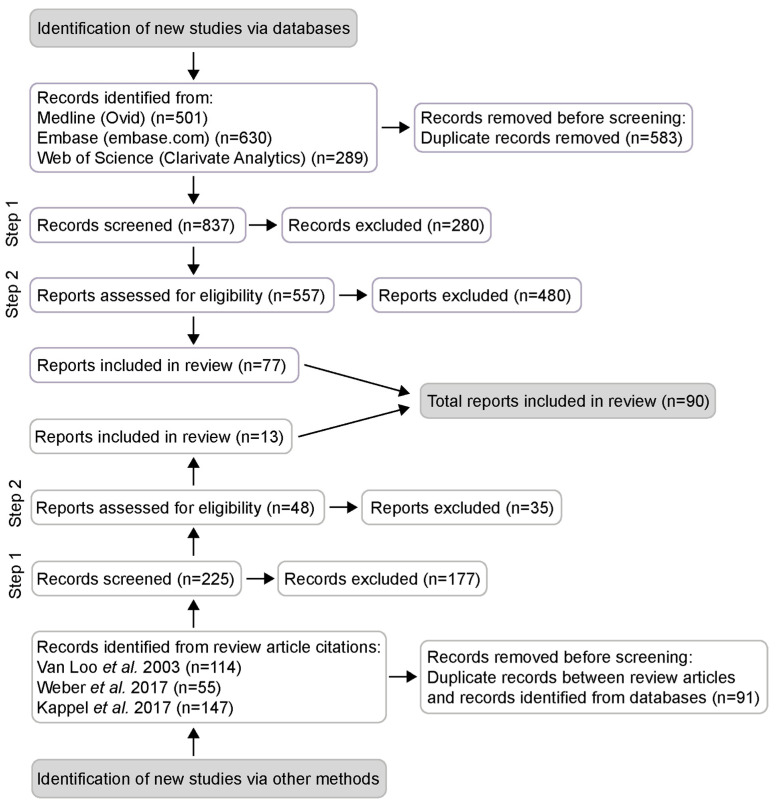
PRISMA flow diagram. Potentially relevant records collected from a systematic literature search using three databases (Medline, Embase and Web of Science) and from scanning the reference lists of three recent literature reviews. Selection performed in two steps; 90 articles suitable for inclusion. Records refer to title, abstract, author keywords and keywords of scientific publications, while reports refer to the full text article.

**Figure 2 animals-13-00143-f002:**
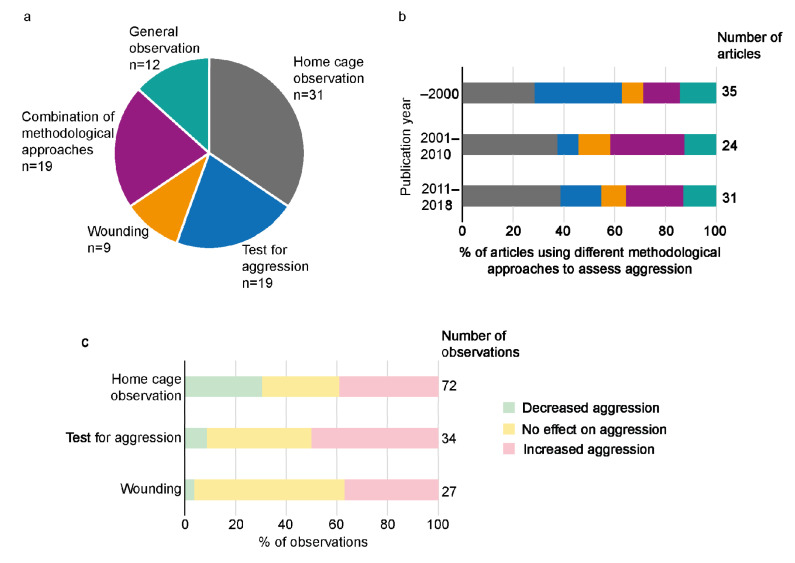
Different methodological approaches to study aggression in the data set. Number of articles (**a**) using the different methodological approaches to assess aggression, and proportion of articles (**b**) using the different methodological approaches before 2000, between 2001 and 2010 and after 2011. The color scheme for the different methodological approaches used in (**a**) also applies to (**b**). The proportion of increased, decreased and no effect on aggression varied between the use of home cage observation, test for aggression and wounding for assessment (**c**). In panel (**c**), observations using more than one methodological approach have been excluded.

**Figure 3 animals-13-00143-f003:**
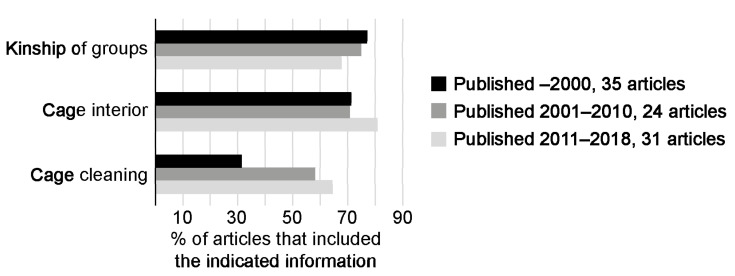
The availability of details on housing and husbandry has changed over time.

**Figure 4 animals-13-00143-f004:**
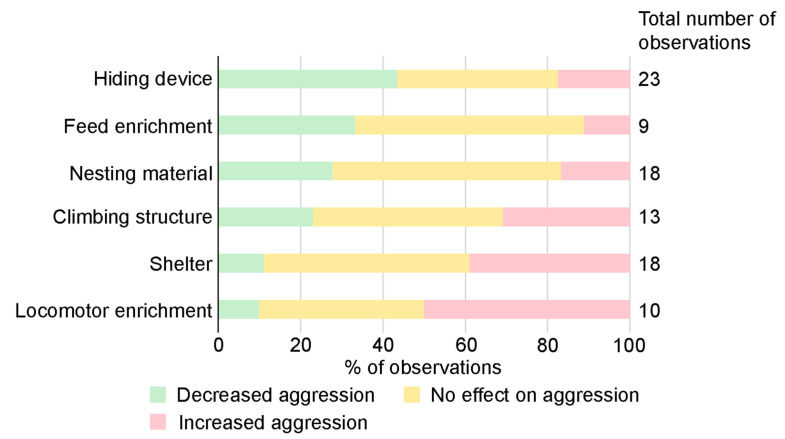
Effect on aggression using different types of enrichment. Note, this includes experiments where either one specific enrichment is used, or when several enrichments are used in combination. In the figure, the category other (11 observations) is excluded due to the variation of enrichments included in that category. Gnawing device is excluded due to a low number of observations (2). Four observations with different outcomes, dependent on the method used to measure aggression, are also excluded.

**Table 1 animals-13-00143-t001:** Treatments used to study the effect on aggression. The different treatments are described in detail in the Methods section. One article can contain observations from more than one treatment category.

Treatments	Number of Articles	Number of Observations
Enrichment	33	64
Time in the group	27	38
Strain	26	35
Group formation	23	30
Housing condition	16	23
Other	7	8

**Table 2 animals-13-00143-t002:** Aggressive behavior, comparing two strains. BALB/c, C57BL/6, and CD-1 were most commonly used in the complete dataset and are therefore color-coded in this table; BALB/c in purple, C57BL/6 in pink and CD-1 in grey.

		Methodological Approach Used to Study Aggression				
Less Aggressive	More Aggressive	Home Cage Behavior	Test For Aggression	Wounding	General Observation	Reference
C57BL/6	BALB/c			x		[75]
C57BL/6	BALB/c	x	x			[73]
C57BL/6	BALB/c	x				[74]
C57BL/6	129S ^a^		x			[23]
C57BL/6	Swiss	x	x			[73]
C57BL/6	NIH	x				[74]
BALB/c	CD-1 ^b^	x		x		[20]
BALB/c	FVB	x			x	[76]
BALB/c	Swiss	x	x			[73]
BALB/c	NIH	x				[74]
BALB/c	C57BL/6	x	x	x		[33]
CBA/J	DBA2		x			[32]
CD-1	BKW	x				[78]
NOD	FVB	x			x	[76]
**No Difference**		**Home Cage** **Behavior**	**Test for** **Aggression**	**Wounding**	**General** **Observation**	**Reference**
C57BL/6	129S ^c^		x			[23]
C57BL/6	BALB/c	x				[30]
C57BL/6	BALB/c	x				[65]
C57BL/6	BTBR	x				[77]
C57BL/6	BTBR	x				[61]
C57BL/6 ^d^	BTBR	x				[77]
C57BL/6	DBA/2		x			[79]
BALB/c	NOD	x			x	[76]

^a^ In non-enriched cages only. ^b^ In severity of wounds. BALB/c had higher number of wounds. ^c^ In cages enriched with a combination of several types of enriched items. ^d^ More chasing observed in C57BL/6.

**Table 3 animals-13-00143-t003:** Articles that studied the effects of group size on aggression. For each study, the group size with least aggression is marked in green. Yellow represents no difference between groups.

	Group Size										Methodological Approach Used to Study Aggression				
Strain	2	3	4	5	6	8	9	10	12	20	Home Cage Behavior	Test for Aggression	Wounding	General Observation	Reference
BALB/c		x		x		x					x		x		[70]
BALB/c		x		x		x								x	[88]
Varying strains	x	x	x	x							x		x		[89]
Randomly bred		x	x	x			x		x		x				[54]
DD/S strain		x		x							x				[59]
MF1		x			x						x				[38]
BALB/c		x					x				x				[92]
CFLP		x			x			x			x				[42]
CFLP		x			x			x			x				[91]
DUB/ICR Swiss	x			x				x			x	x			[90]
DUB/ICR Swiss	x		x			x				x	x	x			[90]
DUB/ICR Swiss	x			x				x			x	x			[90]

## Data Availability

All data generated or analyzed during this study are included in this published article (and its Appendix A).

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
