# Peer review of "Aggression in Group-Housed Male Mice: A Systematic Review"

_animals, 2022, doi:10.3390/ani13010143_

Round 1

Reviewer 1 Report

Dear Authors, 

  I send you the comments on the work titled “Aggression in group housed male mice - a systematic review”.

The introduction and discussion are very well written, the methods used are sound (PRISMA) and the data obtained are very interesting. However, there are a few minor points to be reconsidered:

- You should check the style of the references (in square brackets) and the order (numbering). 

- Introduction.   

Second paragraph: Recommendations on aggression prevention are described and it is said that animals from the same litter can be grouped together before sexual maturity but there are two articles that describe that depending on age male mice from different litters can be grouped together before sexual maturity. One of them even describes, apart from age, factors such as the condition of the cage as well as the number of animals in the receiving group. References: R. Grífols et al. Postweaning grouping as a strategy to reduce singly housed male mice. Animals 2020; doi: 10.3390/ani10112135 and G. Azkona, J. Martín Caballero. Implementing strategies to reduce singly housed male mice. Laboratory Animals. 2019, doi: 10.1177/0023677219845028  

I would rewrite the sentence: These include keeping siblings together or grouping familiar and non-familiar mice before (sentence 54).   

Third paragraph: The social defeat test, the novel arena social encounter test, and the tube test are described in references 2 or 10, and 11?  

- Page 8 line 297, is written all artiklar and I don’t understand the meaning.    - Point 4.5 (page 5, should be 14). You could put the recommendations by points.  

- Use nesting material and other enrichment items that the mice can manipulate and work with. 

- Use hiding devices where the mice can hide. 

Reviewer 2 Report

The present systematic review is a well-done article regarding aggression in mice, an interesting topic for researchers working with laboratory rodents and in the field of animal behavior. I commend the authors on this research since it highlights the important aspects that need to be addressed when trying to evaluate aggression (e.g., the strain, type of enrichment, and the method used). I left some minor comments hoping they can help the authors.

General comment about the citation style: Please, revise the Instruction for authors and amend the citation style throughout the manuscript (e.g., it must be in numerical order inside brackets, the same applies when citing directly the name of the author).

 Lines 53-57: Consider moving this paragraph to line 50, before “However, aggression between male cage mates…). In this way, you can reinforce the recommendations that current guidelines have regarding animal housing, and then state that despite following them, aggression is still observed in mice. Also, could you give one or two examples of strains with a low level of aggression?  

 Lines 80-81: I recommend giving some examples of why the effects of aggression are not clear. For example, some studies where environmental enrichment helped to reduce aggression, mentioned which enrichment was used, and some other studies where the same enrichment (if possible) did not decrease aggression.

 Lines 214-217 and 407-410: I consider this a highly relevant finding of the present article and is well mentioned in the introduction and discussion. Some studies aimed to evaluate aggression take the animals outside of their homes and most of them report increased aggression. However, moving the mice, per se, is an event that might influence their behavioral response. For further articles, it would be interesting to discuss to what extent this influences the reported aggression when exposed to house changes.

Line 259-260: Did the articles that used non-sibling mice or sub-strains of BALB/c mice state why they choose specifically this type of mice? It would be interesting if they reported this data since, according to your results, more than half of the observations were made in these mice and they become aggressive with time, and if this is a general response to the strain, this might influence their results.

 Lines 297-298: Please, revise if the words “alla antiklar” belong to the text.

Line 542-543: I think that another important aspect that could be added (if possible) is if the methods, the evaluator, and their experience in assessing animal behavior have influenced the outcome regarding aggression. Several articles have mentioned the importance of evaluator skills in recognizing specific traits of the species and the state they are evaluating to avoid inaccurate results. 
